# Atomistic modeling of liquid-liquid phase equilibrium explains dependence of critical temperature on γ-crystallin sequence

Sanbo Qin[1] & Huan-Xiang Zhou [1,2✉]

Liquid-liquid phase separation of protein solutions has regained heightened attention for its biological importance and pathogenic relevance. Coarse-grained models are limited when explaining residue-level effects on phase equilibrium. Here we report phase diagrams for γ-crystallins using atomistic modeling. The calculations were made possible by combining our FMAP method for computing chemical potentials and Brownian dynamics simulations for configurational sampling of dense protein solutions, yielding the binodal and critic temperature ($T_c$). We obtain a higher $T_c$ for a known high-$T_c$ γ-crystallin, γF, than for a low-$T_c$ paralog, γB. The difference in $T_c$ is corroborated by a gap in second virial coefficient. Decomposition of inter-protein interactions reveals one amino-acid substitution between γB and γF, from Ser to Trp at position 130, as the major contributor to the difference in $T_c$. This type of analysis enables us to link phase equilibrium to amino-acid sequence and to design mutations for altering phase equilibrium.

[1] Department of Chemistry, University of Illinois Chicago, Chicago, IL 60607, USA. [2] Department of Physics, University of Illinois Chicago, Chicago, IL 60607, USA. ✉email: hzhou43@uic.edu

iquid–liquid phase separation of proteins has been studied intensively in recent years, because the resulting biomolecular condensates both mediate a host of cellular processes ranging from transcription to stress regulation and also are prone to disease-linked aggregation[1–3]. Most of the attention has been focused on intrinsically disordered proteins (IDPs) or proteins with long disordered regions, in part due to the strong tendency of such proteins to undergo phase separation[4–11]. This tendency arises from the fact that intrinsic disorder allows the proteins to easily form multivalent interactions that drive phase separation[12]. Ironically, phase separation was first observed on structured proteins, including γ-crystallins[13,14], which are present in the eye lens of animals at high concentrations. For structured proteins, the formation of multivalent interactions that drive phase separation requires high concentrations[12]. Consequently the phase separation of structured proteins is characterized by a high saturation concentration and/or a low critical temperature ($T_c$), both of which pose difficulties for experimental studies. Whereas the sequence determinants for the phase separation of disordered proteins are well studied[5,7,10,11], our understanding of this crucial issue lags for structured proteins.

Up to six highly homologous γ-crystallins each have been identified in bovine, rat, and human lenses[15–17] (Fig. 1a and Supplementary Fig. 1a). According to their $T_c$ values for phase separation, they can be divided into two groups: one, represented by bovine γB, has $T_c$ below 10 °C; the other, represented by bovine γF, has $T_c$ around body temperature[14,18,19]. High-$T_c$ γ-crystallins are present at a higher level in the lens nucleus than in the cortex, contributing to the gradient in refractive index[14,18]. Phase separation of γ-crystallins may lead to cataract, and is suppressed by other components, such as β-crystallins[20]. When a rat lens was cooled, phase separation was observed, in a phenomenon known as cold cataract[21]. The effects of several cataract-causing point mutations on phase separation have been studied; the resulting changes in $T_c$, e.g., by R14C (in the presence of a reducing agent to prevent disulfide crosslinking) and E107A of human γD were small[22,23].

Given the 82% sequence identity between bovine γB and γF (Fig. 1a), the large gap between their $T_c$ values is astounding and prompted the question about its sequence determinants[14]. The first structural comparison between these two proteins[24] also showed high similarity [Fig. 1b; 0.19 Å root-mean-square-deviation (RMSD) for all backbone atoms]; the authors suggested differences in crystal contact and loop (residues 116–122) conformation as possible causes for the $T_c$ gap. Based on sequence comparison, Broide et al.[19] proposed the amino acid substitutions at positions 22 and 47 as potential, 15 as possible, and 163 as less likely determinants. Norledge et al.[15] noted that these previous attempts were hampered by inaccurate sequences, because the γ-crystallins studied were isolated from bovine lenses and collected as different fractions, and their sequences were not verified. They attributed the $T_c$ gap to differences in overall surface charges, including a higher Arg/Lys ratio and a higher His content in the high-$T_c$ group. However, this attempt had a handicap of its own, i.e., the classification of human and rat γD as high-$T_c$; subsequent studies have shown that both human and rat γD, just like bovine γD[19], belong to the low-$T_c$ group[22,25]. Augusteyn et al.[26] measured the tryptophan fluorescence quenching of different bovine γ-crystallins fractions. Commenting on the latter study, Norledge et al.[15] noted that "their division of the bovine γ-crystallins into two groups on the basis of the tryptophan fluorescence coincides with the division into two groups on the basis of phase separation behavior," and attributed the ready fluorescence quenching of the high-$T_c$ group to an exposed Trp residue, at position 130 (Fig. 1b). Yet, they did not link Trp130 with high $T_c$.

The binodals, i.e., the temperature dependence of protein concentrations in the coexisting dilute and dense phases, measured in ref. [19] for bovine γ-crystallins have motivated a number of computational studies. Dorsaz et al.[27] used spherical particles interacting via a square-well potential to model γ-crystallins. Kastelic et al.[28] extended such a model to have interaction sites located on the spherical surface. Coarse-grained models are becoming successful in reproducing the effects of large sequence variations (e.g., substitutions of all Tyr residues by Phe) on the phase equilibria of IDPs[6,8,9], but quantitative prediction for the effect of a point mutation seems beyond the reach of these models, especially for structured proteins. Recognizing the

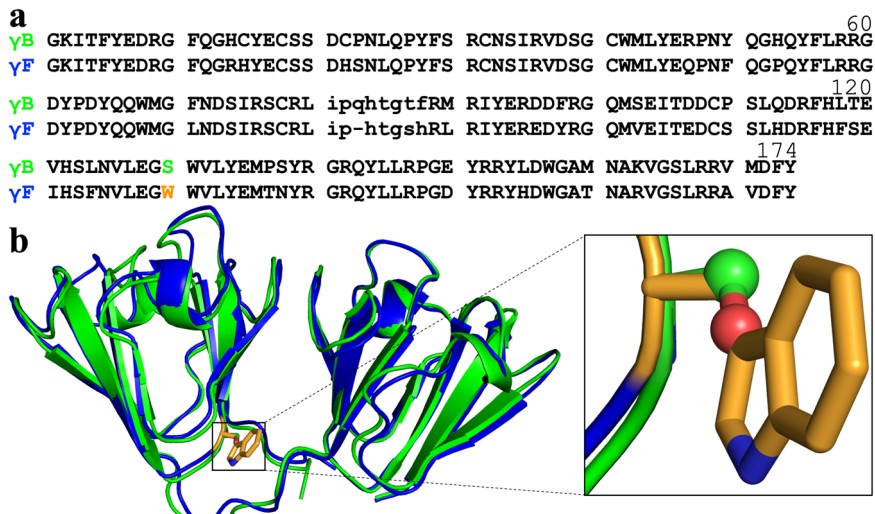

**Fig. 1 Sequence and structure comparison between bovine γB and γF. a** Sequence alignment of γB and γF. Residue numbers are according to γB. The difference at position 130 is highlighted by green and orange colors for the amino acids, S and W, in γB and γF, respectively. The linker between the two domains is in lowercase. The sequences were retrieved from UniProt (https://www.uniprot.org/) entries P02526 and P23005 for γB and γF, respectively, and aligned using ClustalW[58]. The alignment of all known sequences of bovine, human, and rat γ-crystallins is shown in Supplementary Fig. 1a. **b** Superposition of γB and γF crystal structures (PDB entries 1AMM and 1A45). γB and γF are shown in cartoon representation in green and blue, respectively; side chains at position 130 are shown as ball-and-stick for γB and as stick for γF.

limitation of these simplified models, we developed a method called FMAP, or fast Fourier transform (FFT)-based modeling of atomistic protein-protein interactions, and demonstrated its feasibility for studying phase separation for γ-crystallins and other proteins[29] (see Supplementary Note 1 and Supplementary Fig. 2). The power of FFT enables FMAP to handle a vast amount of computation required for determining the binodal of an atomistic protein. The FFT-based approach has also been used to compute the second virial coefficients[30], which are an indicator of the critical temperature[31].

Here we combine FMAP with Brownian dynamics simulations for configurational sampling of dense protein solutions to determine the binodals of bovine γB and γF and identify the origin of their $T_c$ gap. Our residue-specific decomposition of inter-protein interactions and sequence analysis reveal the substitution from Ser in γB to Trp in γF at position 130 (Fig. 1b) as the major contributor to the $T_c$ gap. This work demonstrates a computational approach to map phase equilibrium to amino-acid sequence for structured proteins and opens the possibility for designing mutations to alter phase equilibrium.

## Results

**FFT-based method, FMAPμ, enables calculation of the chemical potentials of γ-crystallins.** The chemical potential $\mu$ is the free energy on a per-molecule basis. For a protein solution, the dependence of the chemical potential on protein concentration ($\rho$) allows the determination of the binodal[12,32]. $\mu$ can be decomposed into

$$\mu = \mu_{\text{id}} + \mu_{\text{ex}} \tag{1}$$

The ideal part $\mu_{\text{id}}$ is the chemical potential of an ideal solution, where the protein molecules have no interactions at all, and is the same as the counterpart of an ideal gas,

$$\mu_{\text{id}} = k_B T \ln(\rho/\rho_0) \tag{2}$$

where $k_B$ is Boltzmann's constant, $T$ is absolute temperature, and $\rho_0$ is an arbitrary reference concentration. The excess part $\mu_{\text{ex}}$ accounts for interactions between the protein molecules in the solution. While low concentration decreases $\mu_{\text{id}}$ (as dictated by Eq. [2]), intermolecular interactions at high protein concentrations can decrease $\mu_{\text{ex}}$. Two coexisting phases must have the same chemical potential. So the simple reason that a protein solution undergoes phase separation is that $\mu_{\text{id}}$ favors the dilute phase whereas $\mu_{\text{ex}}$ favors the dense phase, leading to equality in chemical potential between the phases.

One way to formulate $\mu_{\text{ex}}$ is through the introduction of a test protein, which is identical to the protein molecules in the solution, including experiencing the same intermolecular interactions, but does not affect the protein solution in any way. If we treat each protein molecule as rigid, then the position and orientation of the protein molecule allow the positions of all its atoms to be specified. Let $\mathbf{R}$ and $\boldsymbol{\Omega}$, respectively, be the position and orientation of the test protein, $\mathbf{X}$ be the configurations of the protein molecules in the solution, and $U(\mathbf{X}, \boldsymbol{\Omega}, \mathbf{R})$ be the interaction energy between the test protein and the protein solution, then the excess chemical potential is given by[33]

$$e^{-\beta\mu_{\text{ex}}} = \left\langle e^{-\beta U(\mathbf{X}, \boldsymbol{\Omega}, \mathbf{R})} \right\rangle \tag{3}$$

where $\beta = 1/k_B T$ and $\langle \cdots \rangle$ signifies averaging over $\mathbf{X}$, $\boldsymbol{\Omega}$, and $\mathbf{R}$.

For proteins modeled at the atomistic level, the evaluation of $\mu_{\text{ex}}$ requires a vast amount of computation, involving sampling over $\mathbf{X}$, $\boldsymbol{\Omega}$, and $\mathbf{R}$. For example, the sampling over $\mathbf{R}$ entails placing the test protein (with a particular orientation $\boldsymbol{\Omega}$) into numerous positions inside a protein solution (with a particular configuration $\mathbf{X}$), as illustrated in Fig. 2a. Our FFT-based method,

here termed FMAPμ, accelerates the averaging over $\mathbf{R}$ and thereby makes the evaluation of $\mu_{\text{ex}}$ feasible[29]. Our interaction energy function is comprised of additive contributions from all pairs of atoms between any two protein molecules. The contribution from each pair of atoms $i$ and $j$ has three terms[29,34]:

$$u_{ij}(r_{ij}) = u_{ij}^{\text{st}}(r_{ij}) + s_1 u_{ij}^{\text{n-a}}(r_{ij}) + s_2 u_{ij}^{\text{elec}}(r_{ij}) \tag{4}$$

where $r_{ij}$ is the interatomic distance. The steric term is

$$u_{ij}^{\text{st}}(r_{ij}) = \begin{cases} \infty, & r_{ij} < (\sigma_{ii} + \sigma_{jj})/2 \\ 0, & \text{otherwise} \end{cases} \tag{5}$$

where $\sigma_{ii}/2$ denotes the hard-core radius of atom $i$. The nonpolar attraction term, including van der Waals and hydrophobic interactions, has the form of a Lennard-Jones potential,

$$u_{ij}^{\text{n-a}}(r_{ij}) = 4\epsilon_{ij}\left[\left(\frac{\sigma_{ij}}{r_{ij}}\right)^{12} - \left(\frac{\sigma_{ij}}{r_{ij}}\right)^{6}\right] \tag{6}$$

where $\epsilon_{ij}$ denotes the magnitude of the $i - j$ attraction, $\sigma_{ij}$ is the distance at which the nonpolar interaction energy is zero. As in our previous studies[29,34], we used AMBER parameters for $\epsilon_{ij}$ and $\sigma_{ij}$, along with a scaling parameter $s_1 = 0.16$. The electrostatic term has the form of a Debye-Hückel potential,

$$u_{ij}^{\text{elec}}(r_{ij}) = 332 \sum_{ij} \frac{q_i q_j}{\varepsilon r_{ij}} e^{-\kappa r_{ij}} \tag{7}$$

where $q_i$ is the partial charge on atom $i$, $\varepsilon$ is the dielectric constant, and $\kappa$ is the Debye screening parameter. We introduced a scaling factor $s_2 = 1.6$ to account for effects such as possible reduction in the dielectric constant in dense protein solutions[29].

The sampling of protein configurations ($\mathbf{X}$) was implemented by Brownian dynamics (BD) simulations[35]. 30–450 γB or γF molecules were started in a cubic box of side length 324 Å, resulting in concentrations from 31 to 461 mg/ml that span the experimentally relevant range (see Fig. 3f inset below). The pair distribution function calculated from the BD simulations matches with that expected from the pair interaction energy (Supplementary Fig. 3). From these simulations, we took 2000 configurations at 10-ns intervals to implement FMAPμ. To sample $\boldsymbol{\Omega}$, we generated 500 random orientations for the test protein. Lastly the sampling over $\mathbf{R}$ was realized by discretizing the BD simulation box with a 0.6-Å resolution in each dimension, resulting in a total of $540^3 = 1.57464 \times 10^8$ grid points. So altogether we obtained $1.57464 \times 10^{14}$ interaction energies to calculate $\mu_{\text{ex}}$ at each protein concentration.

In Fig. 2b, c, we display the $\mu_{\text{ex}}$ results for γB and γF over the aforementioned concentration range and at temperatures from $-2\,°C$ to $50\,°C$. Several features at low temperatures are worth noting. First, attractive interactions dominate the Boltzmann average of Eq. [3], and hence $\mu_{\text{ex}}$ is negative. Second and most interestingly, γF $\mu_{\text{ex}}$ is more negative than γB $\mu_{\text{ex}}$. For example, at $T = -2\,°C$ and $\rho = 400$ mg/ml, $\beta\mu_{\text{ex}}$ is $-2.3 \pm 0.1$ for γB but decreases to $-3.0 \pm 0.2$ for γF. The difference in $\mu_{\text{ex}}$ reflects stronger attractive interactions in the γF solution, and translates into a higher $T_c$. So the raw $\mu_{\text{ex}}$ results already capture the key difference between γB and γF in phase equilibrium.

Third, $\mu_{\text{ex}}$ initially decreases with increasing concentration, because more and more molecules can participate in attractive interactions. For example, for γB at $T = -2\,°C$, $\beta\mu_{\text{ex}}$ decreases from $-1.27 \pm 0.02$ at 123 mg/ml to $-2.3 \pm 0.1$ at 400 mg/ml. However, with further increase in concentration, $\mu_{\text{ex}}$ starts to turn over, as molecules experience steric repulsion. Lastly, at the lowest temperatures, the values of $\mu_{\text{ex}}$ can suffer large uncertainties, as illustrated by the $\beta\mu_{\text{ex}}$ value $-4.7$ $(+2.6/-0.7)$ of γB at $-2\,°C$ and 277 mg/ml. The numerical uncertainties arise from the general

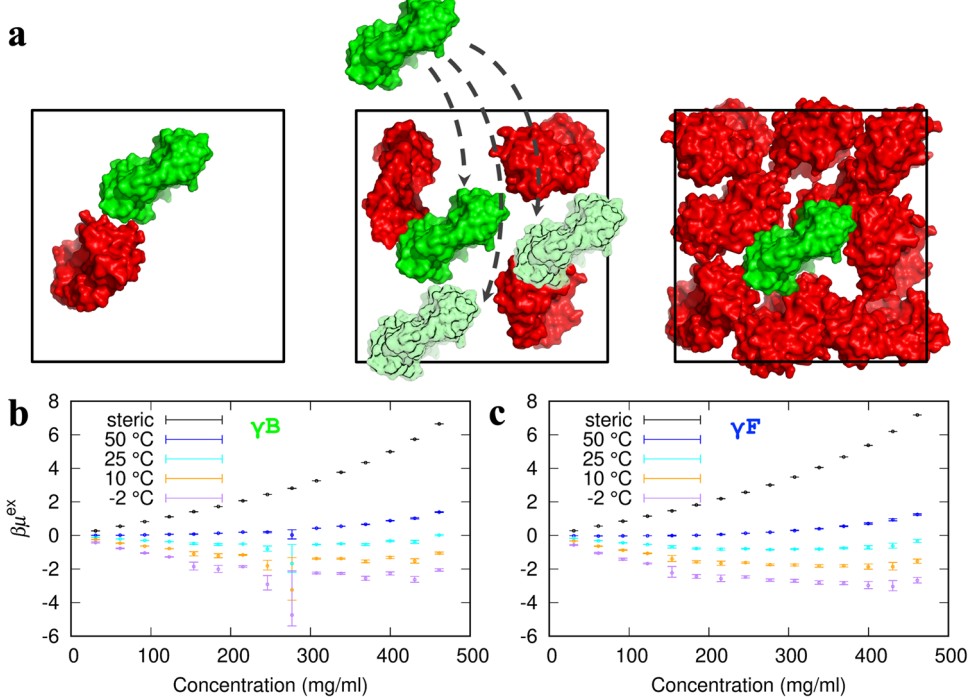

**Fig. 2 FMAPμ calculations and the resulting excess chemical potentials over a range of protein concentrations. a** Illustration of the FMAPμ method. Three boxes with different copy numbers of the protein shown in red depict a range of concentrations. At each concentration, an additional copy, or test protein, shown in green, is fictiously inserted (indicated by dashed arrows) into the box, and its interaction energy with the red copies is calculated by FMAP. The average of the corresponding Boltzmann factor yields the excess chemical potential at that protein concentration. **b, c** Dependences of excess chemical potentials on protein concentration and temperature for γB and γF. Error bars were estimated by applying the blocking method of ref. [59] to the average Boltzmann factors at 500 test-protein orientations. The large error of the γB excess chemical potential at 277 mg/ml and −2 °C was due to a single outlying low interaction energy (see also Supplementary Fig. 5). The steric component is the high-temperature limit of the excess chemical potential. A comparison of the steric components between γB and γF is shown in Supplementary Fig. 4.

difficulty of sampling the lowest energy region of any molecular system (e.g., ref. [4]).

As the temperature is raised, steric interactions become important even at lower protein concentrations, thereby elevating the entire $\mu_{ex}$ curve. At $T = 50$ °C, $\mu_{ex}$ is positive even at the lowest concentration 31 mg/ml for γB and first becomes positive at 184 mg/ml for γF. In this situation, true for all temperatures above $T_c$, no dense phase can achieve equal stability as a would-be dilute phase and thus no phase separation is possible. At the limit of $T \rightarrow \infty$, only the steric term survives in the Boltzmann average of Eq. [3]; the resulting excess chemical potential, $\mu_{ex}^{st}$, is related to the fraction, $f_{CF}$, of test-protein placements that are free of steric clashes with protein molecules in the solution. This relation is given by Eq. [9] in Computational Methods. The clash-free fraction can be obtained accurately at any temperature. As shown in Supplementary Fig. 4, the $f_{CF}$ results of γB and γF are very close to each other, and, at high concentrations, are orders-of-magnitude higher than the counterpart obtained by replacing Lennard-Jones particles with γB molecules, as we did in 2016[29]. The BD simulations here, with an atomistic energy function, capture the ability of protein molecules to form intricate clusters stabilized by preferential interactions, leaving more voids to place the test protein. The excessively low $f_{CF}$, along with limited sampling in $\Omega$, in our 2016 study is responsible for the narrow binodal obtained there.

**Binodals determined from concentration dependence of chemical potential show higher $T_c$ for γF than for γB.** We developed a procedure to mitigate numerical uncertainties of $\mu_{ex}$

manifested at low temperatures. This procedure is based on the observation that the cumulative distribution function (CDF) of the interaction energy $U(\mathbf{X}, \mathbf{\Omega}, \mathbf{R})$ is exponential for over 10 orders of magnitude, covering nearly the entire negative range of $U$ (Supplementary Fig. 5a, b). The procedure involves fitting the logarithm of the CDF to a linear function of $U$ in an intermediate region of $U$ (Supplementary Fig. 5a, b) and then extrapolating to a minimum energy $U_{min}$ (Supplementary Figs. 5a, b and 6[36]). As illustrated in Fig. 3a, the corrected $\mu_{ex}$, by modeling the CDF in the low interaction energy region, is now free of the large variations of the original $\mu_{ex}$.

We then add the ideal part to the corrected $\mu_{ex}$ and perform an equal-area construction[29,32] (Fig. 3b) to obtain the concentrations in the coexisting dilute and dense phases. By connecting the coexisting concentrations from different temperatures, we obtain the binodal (Fig. 3c). The equal-area construction also yields the spinodal, which defines a range of concentrations in which the system is thermodynamically unstable. As shown in Fig. 3c, the computed binodal and spinodal for γB both match well with the experimental counterparts from ref. [37]. The computation yields a critical temperature of approximately 4 °C for γB.

At $T = -2$ °C, the computed spinodal concentrations for γB are 123 mg/ml at the lower end and 400 mg/ml at the higher end. In Fig. 3d, we display a slice of a configuration of the γB solution at these concentrations. Both illustrate the intricate clusters noted above, which again are stabilized by preferential interactions. At the higher concentration, the average number of interaction partners per molecule is higher, leading to a more negative $\mu_{ex}$ as presented above.

In Fig. 3e, we compare the original and corrected $\mu_{ex}$ for γF at −2 °C, along with the corresponding results for γB that have

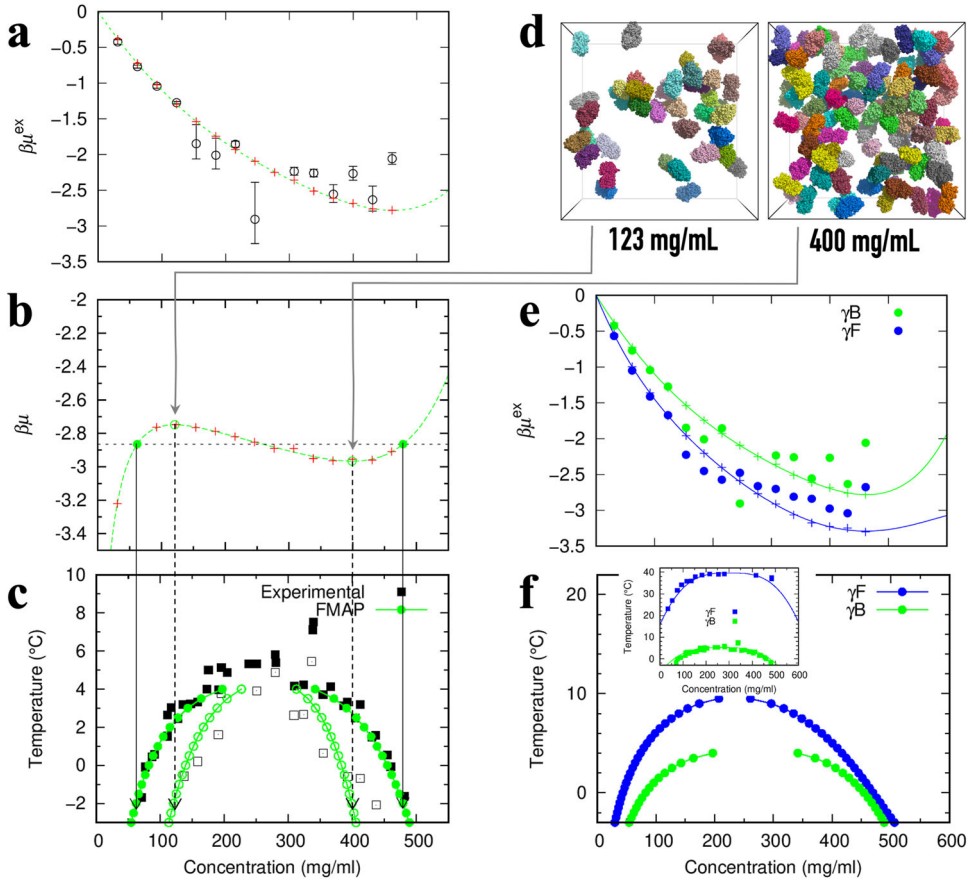

**Fig. 3 Determination of the binodal and spinodal. a** The concentration dependence of the γB excess chemical potential at −2 °C. Circles with error bars are the same raw data shown in Fig. 2b; plus signs display the results after modeling the cumulative distribution function in the low interaction energy region (see Supplementary Fig. 5a, b for illustration); the curve is a fit to a fifth-order polynomial. **b** The total chemical potential, after adding the ideal part. The horizontal line bisects the total chemical potential with equal areas above and below; the first and last intersections define the binodal whereas the extrema define the spinodal. **c** Binodal and spinodal of γB determined by FMAPμ calculations, displayed as filled and open circles, respectively. The solid vertical arrows connect the first and last intersections in (**b**) with the binodal concentrations at −2 °C; the dashed vertical arrows connect the extrema of the total chemical potential with the spinodal concentrations at −2 °C. The experimental binodal and spinodal (from ref. [37] with the concentration correction of ref. [19]) are shown as filled and open squares. **d** Snapshots of γB from BD simulations at the spinodal concentrations (123 and 400 mg/ml). For clarity, only protein copies in a slab spanning one quarter of the side length along z (directed into the page) are displayed. **e** The γB results from panel (a) shown here again in comparison to the counterparts of γF. **f** Comparison of the binodals of γB and γF determined by FMAPμ calculations. Inset: experimental binodals from ref. [19].

already been shown in Fig. 3a. Again, $\mu_{ex}$ is more negative for γF than for γB. Finally, in Fig. 3f, we present the computed binodal for γF, which exhibits a higher $T_c$, ~10 °C, relative to the counterpart for γB. Similar increases in $T_c$ from γB to γF are predicted when other acceptable values of the scaling factors $s_1$ and $s_2$ are applied in the energy function (Eq. [4]) (Supplementary Note 1 and Supplementary Fig. 7). Though moving in the correct direction relative to the γB $T_c$, the predicted γF $T_c$ significantly underestimates the experimental value of ~39 °C[19].

**A more negative second virial coefficient for γF corroborates its higher $T_c$.** Whereas the excess chemical potential $\mu_{ex}$ is determined by the interactions among all the protein molecules in a solution, the second virial coefficient $B_2$ is determined by the interaction between a pair of protein molecules. When $\mu_{ex}$ is expanded as a Taylor series in $\rho$ (see Eq. [19] in Computational Methods), the first-order coefficient is twice of $B_2$. $B_2$ can be an indicator of the critical temperature[31]. A more negative $B_2$ corresponds to a higher $T_c$.

We have adapted the FFT-based approach to derive FMAPB2 as a very robust method for computing second virial

coefficients[30]. The resulting $B_2$ values of γB and γF over a wide temperature range are shown as solid curves in Fig. 4a. It is clear that γF has a more negative $B_2$, therefore supporting a higher $T_c$. The computed $B_2$ values for γB match well with the experimental data of ref. [38] (Fig. 4a inset).

Vliegenthart and Lekkerkerker[31] found an empirical rule, $B_2(T_c)/V_{st} \approx -6$, for spherical particles with a steric core and attractive rim, where $V_{st}$ denotes the volume of the steric core. We can use the steric version of $B_2$, $B_2^{st}$, obtained when only the steric term is kept in the interaction energy function, to define the steric volume: $V_{st} \equiv B_2^{st}/4$. The critical temperatures predicted by the Vliegenthart-Lekkerkerker rule are 0 °C for γB and 6 °C for γF, resulting in the same gap in $T_c$ as determined from the binodals (Fig. 3f). The virial coefficients determined by FMAPB2 are also close to those, displayed as circles, derived from the first-order coefficient of a truncated Taylor expansion of $\mu_{ex}$ (Eq. [19]).

**Residue-specific decomposition of pair interaction energy reveals the S130W substitution as the main cause for $T_c$ difference.** To identify the γB-to-γF substitutions responsible for the increase in $T_c$, we decomposed the pair interaction energies of γB

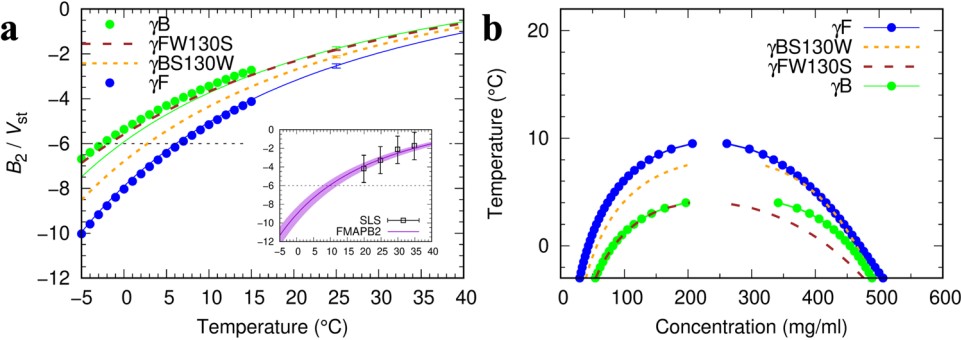

**Fig. 4 Second virial coefficients and binodals of γB, γF, and their point mutants. a** Temperature dependences of $B_2$ for γB, γF, γBS130W, and γFW130S. $B_2$ is expressed in units of the steric volume ($V_{st}$) of the respective protein; $V_{st} = 0.53 \times 10^{-4}$ mol ml/g$^3$ for γB and has very similar values for all γ-crystallins. For γB and γF, circles are from the first-order coefficient of the polynomial fit (Fig. 3a, e); solid curves with matching colors are calculated using FMAPB2[30] (error bars at 25 °C by the blocking method). Results for γFW130S and γBS130W are shown as dashed curves. A horizontal line with an intercept of −6 is drawn according to an empirical rule for determining the critical temperature[31]. Inset: comparison of experimental and computed $B_2$ for γB. Symbols labeled SLS are $B_2$ data obtained by static light scattering in 52.4 mM phosphate buffer (pH 7.1)[38]; a curve with a band displays the FMAPB2 results at the corresponding ionic strength (0.18 ± 0.01 M). **b** Binodals calculated for γFW130S and γBS130W (dashed curves), compared with those for γB and γF (connected circles). The binodal of γFW130S used γB configurations and that of γBS130W used γF configurations.

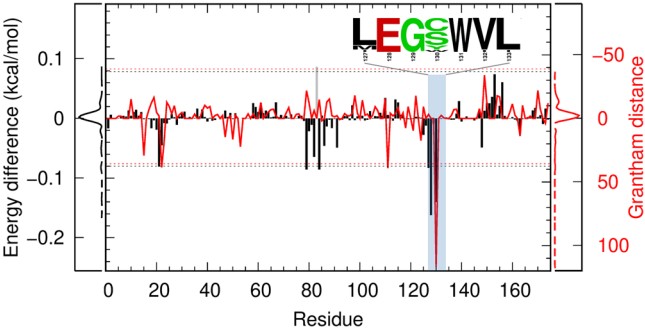

**Fig. 5 Differences in residue-specific contributions to the γB or γF pair interaction energy and residual Grantham distances between low and high-$T_c$ γ-crystallins.** Black bars display the differences between γF and γB in residue-specific decomposed interaction energies; the distribution function of the energetic differences is shown next to the left ordinate. The red curve displays the residual Grantham distances; their distribution function is shown next to the right ordinate. The dotted horizontal lines indicate ± 3 times the standard deviations from the mean, for both residue-specific energetic differences and residual Grantham distances. The inset displays the sequence logo for residues 127 to 133, calculated using the sequence alignment of Supplementary Fig. 1. Note that residue Gln83 of γB aligns to a gap in γF (see Fig. 1). The energetic contribution of γF position 83 was taken to be 0 and the difference calculated by subtracting the γB counterpart is displayed as a gray bar. The residual Grantham distance at this position was taken to be 0.

or γF molecules. Such decomposition has been used previously to reveal residues important for phase separation[39]. In Fig. 5, we display the differences between corresponding residues of γF and γB in their contributions to the respective pair interaction energies. The six positions that make the greatest contributions to the lower pair interaction energy of γF are, in descending order: 128, 130, 84, 79, 127, and 21. The differences in energetic contributions at all these positions are beyond the cutoff set at mean − $3 \times$ SD. γB and γF have the same amino acids at five of these positions: Asp21, Arg79, His84, Leu127, and Glu128. The only position that involves a change in amino acids is 130, from Ser in γB to Trp in γF. Therefore the decomposition reveals the S130W substitution as the main cause for strengthening intermolecular interactions, leading to the higher $T_c$ of γF.

To verify the role of the S130W substitution, we modeled the W-to-S mutation in γF and the reverse mutation in γB, and calculated their effects. As shown in Fig. 4a, the γFW130S mutant largely mimics γB in $B_2$, with only slight undershooting of its magnitude at low temperatures. The γBS130W mutation partially recovers the large negative $B_2$ of γF. Similar descriptions apply to the effects of the two mutations on residue-specific decomposed pair interaction energies. As shown in Supplementary Fig. 8a, the γF − γFW130S differences in residue-specific contribution are similar to the γF–γB counterparts (Fig. 5). In particular, at Asp21 and Glu128 along with position 130, the γF–γFW130S differences exceed the mean − $3 \times$ SD cutoff. On the other hand, the γBS130W mutant does not quite recover the pair interaction energy of γF; only the γBS130W–γB difference in residue-specific contribution at position 130 exceeds the mean − $3 \times$ SD cutoff (Supplementary Fig. 8b).

As the ultimate test of the role of the S130W substitution, we determined the binodals of the γFW130S and γBS130W mutants (Fig. 4b). The binodal of the γFW130S mutant is close to that of γB, while the binodal of the γBS130W mutant comes close to that of γF.

**γB interactions are more isotropic whereas γF interactions more preferentially involve Trp130.** To provide further insight into the S130W substitution, we analyzed the poses with the lowest pair interaction energies from FMAPB2 calculations. We label one partner of the pair as the central copy and the other partner as the test protein. The poses of the low-energy pairs are displayed in Fig. 6a for γB and 6b for γF. The central copy is rendered in cartoon or surface and the test protein represented by a dot at its center of geometry, one for each low-energy pose. We define a molecular frame, with the $z$ axis parallel to its inter-domain interface and the $y$ axis pointing from the C-terminal domain to the N-terminal domain. The view in Fig. 6a, b is into the negative $x$ axis of the central copy. A 360° view of Fig. 6a, b is presented in Supplementary Movies 1 and 2.

In the top left panels of Fig. 6a, b, we display the 1000 lowest-energy poses. For both γB and γF, the test protein tends to concentrate in positions facing residue 130 of the central copy, but this tendency is stronger for γF. We clustered the 1000 poses and present the large clusters (≥20 members) in the top right panels of Fig. 6a, b. Each cluster is represented by a sphere and an arrow. The sphere has a radius proportional to the cluster

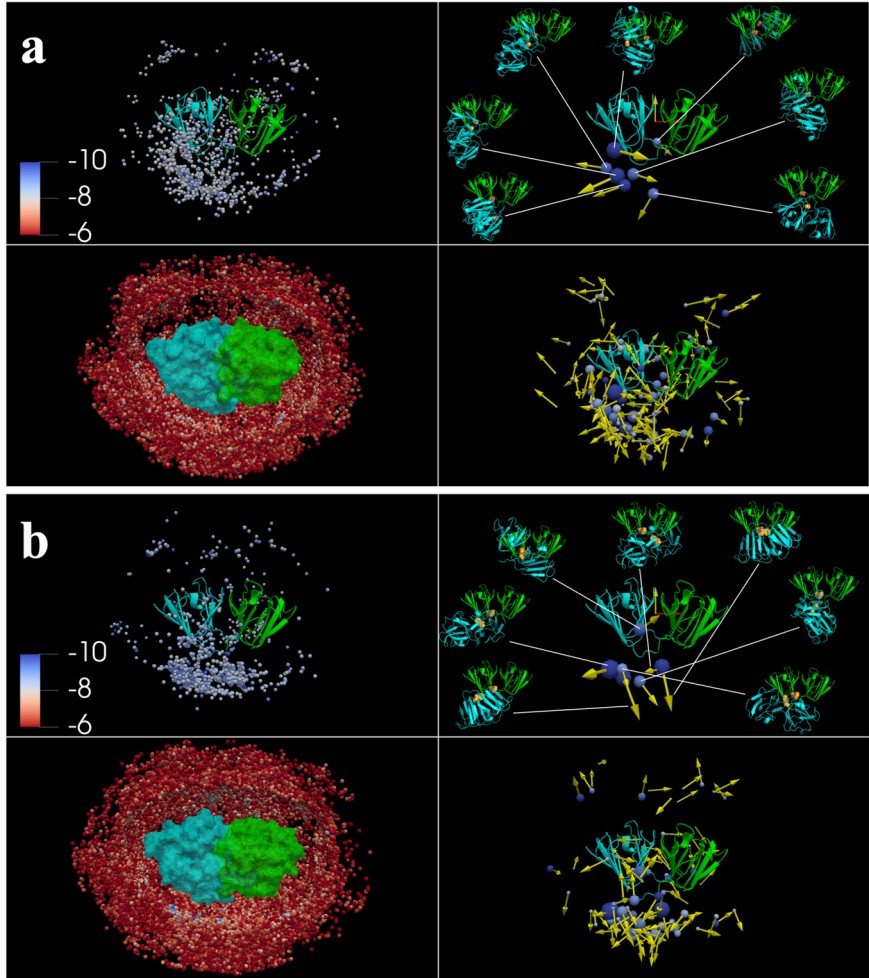

**Fig. 6 Lowest-interaction energy γB or γF pair configurations from FMAPB2 calculations. a** Distribution of lowest-interaction energy configurations for γB. Top left: 1,000 lowest-interaction energy configurations, shown with the central copy in cartoon representation, and the test protein as a dot located at its center of geometry. The dots are colored according to the interaction-energy scale in the inset. Top right: cluster representation of the 1,000 configurations. In the structure of the lowest-energy member of each large cluster, the Ser130 sidechain is shown in sphere representation. Bottom right: same as the top right panel except that the threshold cluster size is reduced to 2. Bottom left: same as the top left panel except that the selection criterion was an upper bound of -6 kcal/mol for the interaction energy. The central copy is rendered as surface, while dots at the centers of the test protein are shown with a near clipping plane at 15 Å from the center of the central copy. **b** Distribution of lowest-interaction energy configurations for γF, presented in the same way as in (**a**).

size and is centered at the pose with the lowest energy in that cluster; the arrow is along the z axis of the molecular frame of the test protein in the latter pose. Both γB and γF have seven large clusters. For γB, six of the seven clusters are around residue 130 and the seventh cluster is at the back of the central copy; together the seven clusters account for 30% of the 1000 low-energy poses. For γF, all the seven clusters are around residue 130 and collect 47% of the 1000 lowest-energy poses, indicating a much stronger preference for residue 130 to be an interaction site.

We also display the cluster representatives by their molecular structures in the top right panels of Fig. 6a, b. As further illustrated in Supplementary Fig. 9a, the two partner γF molecules prefer a parallel relative orientation, but the γB pairs prefer a perpendicular orientation. Moreover, while γF pairs have a strong tendency to form interfaces that bury the Trp130 residues in both partner molecules, γB Ser130 can be either inside or outside the interfaces of the low-energy pairs.

Trp has the bulkiest sidechain, possessing the potential for hydrophobic, cation–π, amido–π, π–π, and hydrogen bonding interactions. The roles of Trp-mediated cation-π interactions in the folding stability of structured proteins and binding stability of IDPs are well recognized[40,41]. In contrast, Ser has only a short polar sidechain, with interaction limited to hydrogen bonding. In γ-crystallin structures, residue 130 is located at the foot of the inter-domain cleft (Fig. 1b). In interfaces that bury the Trp130 residues in both partners, Trp130 in one partner can interact with surface residues around Trp130 of the other partner, including Asp21, Arg79, Leu127, and Glu128 (Supplementary Fig. 9b). In contrast, for γB, Ser130 has limited ability to form these interactions even when buried in an interface; instead other interactions, such as between Arg153 in one partner and Glu150 of the other partner, stabilize the interface (Supplementary Fig. 9c).

The foregoing structural differences between γB and γF in their respective pairs can now explain the energetic differences in Fig. 5. γF pairs form interfaces where both Trp130 residues are buried; interactions between Trp130 on one side and Asp21, Arg79, His84, Leu127, and Glu128 on the opposite side result in the negative energy values at these residues. γB pairs can form alternative interfaces, which explain the positive energy values at Arg153 and Asp156.

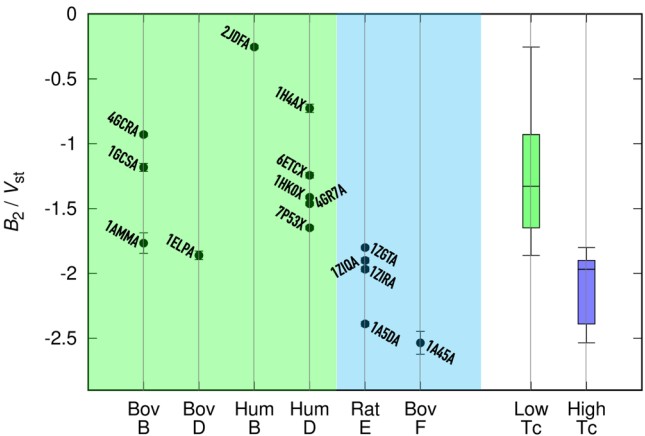

**Fig. 7 The second virial coefficients calculated by FMAPB2 on 15 γ-crystallin structures in the PDB.** The labels are the PDB entry names, with the fifth letter denoting the chain; low and high-$T_c$ entries are displayed in regions shaded green and blue, respectively. Error bars are determined by the blocking method. "Human" and "bovine" are abbreviated as Hum and Bov, respectively. Box plots for the 10 low-$T_c$ $B_2$s and 5 high-$T_c$ $B_2$s are shown at the far right.

While the interfaces in the representatives of the large clusters are energetically favorable, it is important to point out that γB or γF can form numerous other binary interfaces. A fuller picture is presented when all clusters with at least two members are included (bottom right panels of Fig. 6a, b). Poses start to populate the rest of the surface of the central copy. An even fuller picture is reached when all poses with pair energies below -6 kcal/mol are included (bottom left panels of Fig. 6a, b). For γF, although Trp130 is a preferred interaction site, many other sites can also participate in intermolecular interactions, allowing the formation of clusters beyond a dimer in a concentrated solution and thereby leading to phase separation. This scenario is illustrated well using the Trp130-Trp130 distance between two neighboring γF molecules. The distribution of Trp130-Trp130 distances in low-energy pairs, weighted by the Mayer function, exhibits a high peak around 10 Å (Supplementary Fig. 9d), reflecting the Trp130-Trp130 interface noted above. Trp130-Trp130 distances in γF clusters formed in the dense phase (as captured by BD simulations) still show a peak around 10 Å (Supplementary Fig. 9e), indicating that Trp130-Trp130 interfaces are also formed to stabilize the clusters. However, beyond this peak, Trp130-Trp130 distances exhibit a broad distribution, indicating the participation of other interfaces. In comparison, Ser130-Ser130 interfaces have a reduced tendency to form in low-energy γB pairs (Supplementary Fig. 9d) and play a reduced role in γB clusters (Supplementary Fig. 9e).

**Grantham's distance between low and high-$T_c$ γ-crystallins also singles out the S130W substitution.** We calculated Grantham's distance[42] between the low-$T_c$ group (γA, γB, γC, and γD) and the high-$T_c$ group (γE and γF) at each position along the sequence alignment in Supplementary Fig. 1a. Grantham's distance measures the physicochemical difference between the side chains of two amino acids. Its square is a weighted sum of squared differences in three properties: group composition, polarity, and molecular volume. The residual Grantham distance, defined as the difference between inter-group and intra-group distances, is displayed as a red curve in Fig. 5. The values exceed the cutoff of mean $+3 \times$ SD at three positions: 22, 111, and 130.

In Supplementary Fig. 1b, we specifically compare the amino acids of the 15 γ-crystallins at each of these three positions. At

position 22, the low-$T_c$ group contains three unique amino acids: Asn, Cys, and His whereas the high-$T_c$ group contains only His. Therefore the two groups share the amino acid His at position 22. A similar situation occurs at position 111, where the two groups share the amino acid Ser. Only at position 130, the two groups do not share any amino acid: the low-$T_c$ group contains Cys and Ser, whereas the high-$T_c$ group contains Trp and Tyr, which are both aromatic. The analysis using Grantham's distance thus singles out the substitutions at position 130 as involving the greatest physicochemical difference between the low and the high-$T_c$ groups, from a short sidechain to an aromatic sidechain. The sequence logo around position 130 is shown as an inset in Fig. 5, illustrating the divergence at position 130 and consensus at neighboring positions.

**High-$T_c$ γ-crystallins collectively have more negative virial coefficients than low-$T_c$ γ-crystallins.** The energetic differences presented above between γB and γF not only are not limited to the Protein Data Bank (PDB) structures chosen for the calculations but also extend to other low and high-$T_c$ γ-crystallins. We expanded the FMAPB2 calculations to 13 other PDB structures (Fig. 7). Together the 15 structures cover six different γ-crystallins; 10 of these structures are in the low-$T_c$ group and the remaining five are in the high-$T_c$ group. The $B_2/V_{st}$ values are $-1.3 \pm 0.5$ (mean $\pm$ SD) for the low-$T_c$ group and $-2.1 \pm 0.3$ for the high-$T_c$ group. The high-$T_c$ group has a very significantly lower mean $B_2$ ($P$ value $= 0.002$ in unpaired $t$ test), consistent with a much higher $T_c$.

## Discussion

Based on extensive sequential, structural, and energetic analyses, we have identified a Ser to Trp substitution at position 130 as the main cause for the difference in $T_c$ between γB and γF. This identification finally provides a solution to the question first raised by Siezen et al.[14] nearly 40 years ago. More broadly, our study demonstrates that it is now possible to quantitatively account for the contributions of individual residues to the energetics of phase separation for structured proteins. Such quantitative characterization enables us not only to understand how sequence determines phase equilibrium but also to design sequences with the critical temperature altered in a desired direction.

Trp (or Tyr) as an amino acid that promotes phase separation has been well-recognized for IDPs[5,7,10]. Here we show that a substitution from Ser to Trp has a major role in raising the $T_c$ of a structured protein. In IDPs, a Trp can take advantage of their flexibility to form a variety of intermolecular interactions including hydrophobic, cation–π, amido–π, π–π, and hydrogen bonding. In structured proteins, the same potential for intermolecular interactions is open to a Trp when we consider the fact that the dense phase is stabilized by nonspecific intermolecular association that can occur in countless ways.

Whereas the effect of an S-to-W mutation could be context-dependent for IDP, it may be much more so in a structured protein. For example, an S-to-W mutation in a buried site (assuming that the protein structure remains stable), is expected to have little effect on phase separation because the residue does not participate in intermolecular interactions. Moreover, not all exposed S-to-W mutations are created equal. An S-to-W mutation at a site where low-energy pairs are unlikely to form may have only a modest effect, but an S-to-W mutation at a preferential binding site, such as at the foot of the inter-domain cleft of γ-crystallins (Fig. 6), is likely to promote phase separation. The S130P mutation of human γD abolished heat-induced aggregation[43], possibly by promoting dimers that hide the

residue-130 sites in both monomers and thus prevent further aggregation. We conclude that the significant contribution of the S130W substitution to γF $T_c$ derives from not only the physico-chemical differences between the two amino acids but also from the strategic location of residue 130 on the protein surface.

While both IDPs and structured proteins can undergo phase separation, it has been recognized that IDPs more readily do so than structured proteins[12]. In essence, the flexibility of IDPs makes it easier for them to form intermolecular interactions that stabilize the dense phase. In contrast, structured proteins require high concentrations for a given protein molecule to interact with a sufficient number of partner molecules in order to make the dense phase stable. As a result, the dense-phase concentration of a structured protein is much higher than the counterpart of a typical IDP, corresponding to a much broader binodal. One indication of this concentration difference is the much higher dense-phase viscosities of structured or multi-domain proteins relative to IDPs[44]. For γB and γF, the dense-phase concentrations are 400–500 mg/ml, or around 20 mM[19]. Atomistic modeling presented here captures well the countless ways that proteins can form intermolecular interactions in the dense phase and thereby can recapitulate the broad binodals.

By exploiting the power of FFT, our approach has overcome the challenge presented by the vast amount of computation required for determining the binodal of an atomistic protein. Still, the approach has significant room for improvement, including the accuracy of the interaction energy function and the treatment of protein flexibility. These limitations most likely account for the present underestimation of the $T_c$ gap between γF and γB. Binodals are sensitive to energy functions and thus hold great potential for their parameterization. For example, our current energy function may underestimate the distinction between Arg and Lys. A substitution from Lys to Arg at position 163 could contribute to the high $T_c$ of γF[15,19]; such substitutions in Antarctic toothfish γ-crystallins have indeed been found to have a significant effect on $T_c$[45]. Our preliminary calculation using the Rosetta all-atom energy function[46], which is widely used in protein docking and design and more sophisticated than our energy function, shows a wider gap between γF and γB in binary interaction energies (Supplementary Note 1 and Supplementary Fig. 10). As for protein flexibility, an important step forward may be sidechain repacking in response to the approach of other protein molecules. Our preliminary study, by using RosettaDock[47] to carry out sidechain repacking, indeed shows a further widening of the energy gap between γF and γB (Supplementary Note 1 and Supplementary Fig. 10). Alternatively, using a preselected library of protein structures in simulations[48] and for chemical potential calculations also seems a promising direction. As an initial test of this idea, we calculated the $B_2$ value by averaging over multiple structures from the PDB for a given γ-crystallin (Supplementary Note 1 and Supplementary Fig. 11). These average $B_2$ values show the expected gap between the low and high-$T_c$ groups, now with a higher confidence level since it is based on multiple structures.

## Computational methods
**Protein structure preparation**. The structures of the γ-crystallins were from PDB entries 1AMM for bovine γB[49] and 1A45 for bovine γF[15]. PDB2PQR[50] was used to add hydrogens and assign AMBER charges, with protonation states assigned according to PROPKA[51] at pH 7. The net charge is 0 for γB and +1 for γF. Additionally, 13 other high-resolution (2.3 Å or better) γ-crystallin X-ray structures were downloaded from the PDB and similarly processed (see Fig. 7 for PDB names). The S130W mutation of γB was modeled by grafting the sidechain of Trp130

in γF after structure alignment using all backbone atoms. The newly introduced Trp130 sidechain clashed with the nearby Arg147 sidechain, and so these two sidechains were refined by steepest-descent energy minimization in UCSF Chimera[52]. The W130S mutation of γF was similarly modeled. Four of the five human γD structures (other than 1HK0) contained single or double mutations; each of the mutated sidechains was reverted to the wild-type sidechain, by grafting the corresponding sidechain in 1HK0 (after structure alignment using backbone atoms of the residue in question plus two neighboring residues in either direction) and then energy minimization. Unless otherwise indicated, the ionic strength was 0.24 M, modeling 100 mM phosphate buffer at pH 7.1 in experimental studies of γ-crystallin phase separation[19].

**Brownian dynamics (BD) simulations**. The simulation of diffusional association package (SDA, version 7.2.2)[35] was used to generate protein configurations at multiple concentrations. The simulation box was a cube with a side length of 324 Å; periodic boundary conditions were imposed. The number ($N$) of proteins inside the box was 30 or a higher multiple, all the way to 450, covering a concentration range of 31–461 mg/ml. For initial configurations, the protein molecules were oriented randomly and placed at the positions of Lennard-Jones particles from previous simulations[29]. The effective charges derived from solving the Poisson-Boltzmann equation (with van der Waals surface as the dielectric boundary) using APBS[53] were used for electrostatic solvation energy calculation[54]. The temperature was 300 K. The interaction energy between protein molecules consists of Coulomb interaction, electrostatic solvation, hydrophobic solvation, and soft-core repulsion. Default parameter values were used except for the last term, for which we reduced the scaling factor by fourfold (from 0.0156 to 0.0039). The latter value allowed for a good match with FMAPB2 in pair correlation functions (Supplementary Fig. 3).

After 1 μs equilibration, snapshots were collected at 10-ns intervals for the next 20 μs, totaling 2000 snapshots. Depending on the protein number $N$, the BD simulations took 5–330 h on 16 cores in two Intel(R) Xeon(R) CPU E5-2650 2.6 GHz CPUs.

**FMAPμ calculations**. The interaction energy function is specified by Eqs. [4] to [7]. A cutoff of 12 Å was imposed for calculating interactions between two atoms. For a given configuration $\mathbf{X}$ of the protein solution and a given orientation $\boldsymbol{\Omega}$ of the test protein, the interaction energy $U(\mathbf{X}, \boldsymbol{\Omega}, \mathbf{R})$ when placing the position $\mathbf{R}$ of test protein at any of the $540^3 = 1.57464 \times 10^8$ points on a cubic grid was calculated using FMAP[29,32,34]. For each protein concentration, the calculation was repeated $2000 \times 500 = 10^6$ times, combining 2000 protein configurations with 500 orientations of the test protein. These calculations took about 3500 h per protein concentration on 16 cores in two Intel(R) Xeon(R) E5-2650 2.6 GHz CPUs. The excess chemical potential is finally determined according to Eq. [3].

For a given $\mathbf{X}$ and a given $\boldsymbol{\Omega}$, FMAP returns the number of grid points where the test protein experiences steric clash and the interaction energies at all the clash-free grid points. The latter grid points constitute the clash-free fraction, $f_{CF}$. The value of $f_{CF}$ is very stable with respect to the change in $\mathbf{X}$ or $\boldsymbol{\Omega}$[55]; therefore we can treat $f_{CF}$ as a constant and pool the clash-free interaction energies from the $10^6$ combinations of $\mathbf{X}$ and $\boldsymbol{\Omega}$, leading to

$$e^{-\beta\mu_{ex}} = \frac{f_{CF}}{M_{CF}} \sum_{m=1}^{M_{CF}} e^{-\beta U(\mathbf{X}, \boldsymbol{\Omega}, \mathbf{R})} \tag{8}$$

where $M_{CF}$ is the total number of clash-free interaction energies. The maximum possible value of $M_{CF}$ is $1.57464 \times 10^{14}$. We refer

to this formulation of the excess chemical potential as "raw-sum". In the hypothetical limit of $T \to \infty$, we can set $\beta$ to 0 in the right-hand side of Eq. [8] and obtain

$$e^{-\beta \mu_{ex}^{st}} = f_{CF} \tag{9}$$

**Calculating $\mu_{ex}$ over a range of temperatures.** Results for $\mu_{ex}$ over a range of temperatures are needed to determine the binodal. Instead of repeating the above FMAPμ calculations at different temperatures, we calculated the interaction energies once at 298 K and then merely changed $\beta$ when carrying out the sum over the Boltzmann factors (Eq. [8]). This shortcut assumes that the interaction energy function is temperature-independent and that the same BD configurations can be used for FMAPμ calculations at different temperatures.

We did not save the individual interaction energies, which could be up to $1.6 \times 10^{14}$ in total number, but binned them into a histogram in very fine bins (width $\Delta$ at 0.016 kcal/mol). We did save all the configurations with interaction energies lower than $-8$ kcal/mol for further analysis. The raw sum in Eq. [8] can now be expressed as

$$e^{-\beta \mu_{ex}} = \frac{f_{CF}}{M_{CF}} \sum_{U=U_{min}}^{U_{max}} H(U) e^{-\beta U} \tag{10}$$

where $H(U)$ is the number of interaction energies in the bin centered at $U$; and $U_{min}$ and $U_{max}$, respectively, are the minimum and maximum interaction energies. Note that the Boltzmann factor $e^{-\beta U}$ is a decaying function of $U$ whereas $H(U)$ is expected to be a growing function of $U$ as $U$ increases from $U_{min}$. Depending on whether the rate of decay is lower or higher than the rate of growth, the product, $H(U)e^{-\beta U}$, is either a growing or decaying function of $U$. The rate of decay of $e^{-\beta U}$ depends on temperature. For high temperatures, $H(U)e^{-\beta U}$ is a growing function of $U$; hence its values at $U$ near $U_{min}$ contribute little to the sum in Eq. [10] and the inaccuracy in $H(U)$ near $U_{min}$ does not cause serious numerical errors in $\mu_{ex}$. However, at low temperatures, $H(U)e^{-\beta U}$ is a decaying function of $U$; then this inaccuracy in $H(U)$ can result in significant uncertainties in $\mu_{ex}$, as illustrated by the result for γB at 277 mg/ml and $-2\,°C$ (Fig. 2b).

We not only saved the total interaction energies but also the separate terms (i.e., nonpolar attraction and electrostatic) in a histogram form. These data allowed us to quickly produce chemical potentials and other results when we changed the values of the scaling factors $s_1$ and $s_2$.

**Modeling $H(U)$ near $U_{min}$ as a power law distribution with an upper bound.** To mitigate the aforementioned uncertainties in $\mu_{ex}$, we modeled $H(U)$ in the low interaction energy region. We observed that the (unnormalized) cumulative distribution function (CDF)

$$C(U) = \sum_{U' \geq U_{min}}^{U} H(U') \tag{11}$$

has an exponential dependence on the interaction energy (e.g., Supplementary Fig. 5a, b)

$$C(U) = A e^{\alpha U}, \quad U_{min} \leq U < 0 \tag{12}$$

for over 10 orders of magnitude, covering nearly the entire negative range of $U$. The values of $\alpha$ range from 2.1 to 1.5 (decreasing with increasing protein concentration), comparable to the value, 1.7, of $\beta$ at 298 K. Correspondingly, the histogram

function has the form

$$H(U) = \frac{dC(U)}{dU} \tag{13a}$$

$$= A\alpha e^{\alpha U}, \quad U_{min} \leq U < 0 \tag{13b}$$

A variable change

$$x = e^{-\beta U} \tag{14}$$

turns $H(U)$ into a power law distribution:

$$J(x) \equiv H(U) \frac{dU}{dx} \tag{15a}$$

$$= A\left(\frac{\alpha}{\beta}\right) e^{-\left(\frac{\alpha}{\beta}+1\right)x}, \quad 1 < x \leq b \tag{15b}$$

where

$$b = e^{-\beta U_{min}} \tag{16}$$

is the upper bound of $x$ in the power law distribution.

Instead of using an exact power law distribution in $x$, we fit the logarithm of the CDF to a linear function of $U$ using the local linear regression program loess in the R package (http://cran.r-project.org/; with degree = 1 for linear regression and span = 0.75 as the fraction of data points for locally weighted fitting). The fit was limited to a portion of the CDF bounded below by CDF = $10^4$ and above by $U = -4$ kcal/mol (Supplementary Fig. 5a, b). The fit function was then extrapolated to $U = U_{min}$. The determination of $U_{min}$ is described next.

One possible estimate for $U_{min}$ is the observed lowest interaction energy, but such an estimate is subject to significant statistical uncertainties. A more robust method for estimating the upper bound $b$ (equal to the Boltzmann factor of $U_{min}$) in a power law distribution was developed recently[36]. In this method, one calculates the mean value of the largest observed $x$ among replicate samples of a given size and then fits the dependence of the mean largest value on sample size to a function.

To apply this method, we divided the full list of clash-free interaction energies into blocks, each of which was generated from a fixed number ($M_{ori}$) of test-protein orientations. Each block is a replicate sample. The block size $M_{ori}$ is effectively a measure of sample size; the number of interaction energies for a single test-protein orientation is actually up to $2000 \times 1.57464 \times 10^8 = 3.1 \times 10^{11}$. The total number of orientations in our calculations was 500 (denoted as $M_{tot}$); $M_{ori}$ could be any factor of $M_{tot}$, including 1, 2, 4, 5, 10, 20, 25, 50, 100, 125, 250, 500. The number of blocks, or replicate samples, was then $M_{rep} = M_{tot}/M_{ori}$. For each block, the lowest interaction energy was found; the average of the block-specific lowest interaction energies was then evaluated to yield the mean lowest interaction energy, $\widehat{U}_{min}$, for a given block size $M_{ori}$. The dependence of $\widehat{U}_{min}$ on $M_{ori}$ was finally fit to

$$\widehat{U}_{min} = U_{min} + \frac{E}{1 + (M_{ori}/D)^\delta} \tag{17}$$

where $U_{min}$, $D$, $E$, and $\delta$ are fitting parameters, with $\delta$ restricted to [0.6. 1][36]. To reduce variations in the fit values of $U_{min}$, we simultaneously fit the $\widehat{U}_{min}$ data of a protein at different concentrations to Eq. [17] and treated $D$ as a global parameter. Fitting was done by calling the least_squares function in scipy (https://scipy.org/), with the trust region reflective algorithm as the method of minimization. In addition to obtaining $\widehat{U}_{min}$ from the FMAP output, we also took the 50 configurations with the lowest FMAP interaction energies from each test-protein orientation and recalculated the exact interaction energies by an

atom-based method[56], thereby yielding a second set of $\widehat{U}_{\min}$ results. The fitting to Eq. [17] is illustrated in Supplementary Fig. 6. The resulting $U_{\min}$ values are presented in Supplementary Fig. 5c, d, where one can see that the FMAP and atom-based methods gave very similar results.

To further smooth $U_{\min}$ as a function of the protein concentration, we assumed the endpoint of the CDF extrapolation to have the following dependence on the clash-free fraction at the given protein concentration:

$$\text{CDF}(U_{\min}) = Bf_{\text{CF}}{}^{\gamma} \qquad (18)$$

With the coefficient $B$ chosen to be 1 and the exponent $\gamma$ chosen to be 0.25, Eq. [18] closely models the $U_{\min}$ results obtained according to Eq. [17], as shown in Supplementary Fig. 5c, d. So finally we used Eq. [18] to determine $U_{\min}$ values for ending the CDF extrapolation. The resulting $\mu_{\text{ex}}$ is now a smooth function of the protein concentration (e.g., Fig. 3a), free of large variations caused by a single outlying low interaction energy. Such outliers can be identified by a large (>1.0 kcal/mol) deviation from the expected $U$ when the fit function for the CDF is extrapolated to $\text{CDF} = 1$ (Supplementary Fig. 5b inset). The $U_{\min}$ values obtained according to Eq. [17] in cases with such outliers are displayed as open symbols in Supplementary Fig. 5c.

In short, we applied local linear regression to a portion of the CDF (above $\text{CDF} = 10^4$) and then extrapolated down to the lower bound given by Eq. [18]. In calculating $\mu_{\text{ex}}$, the extrapolated CDF was used between the lower bound and $\text{CDF} = 10^4$, but the original CDF was used above $\text{CDF} = 10^4$.

**Determining binodal and spinodal.** Once $\mu_{\text{ex}}$ is calculated as described above for a range of protein concentrations at a given temperature, we fit it to a fifth-order polynomial[29]:

$$\beta\mu_{\text{ex}} = \sum_{l=1}^{5} b_l \rho^l \qquad (19)$$

This fit makes it easier to determine the binodal and spinodal. Equation [19] is a truncated Taylor expansion; in a full expansion (related to the virial expansion of the osmotic pressure), the first-order coefficient, $b_1$, is twice the second virial coefficient $B_2$. Adding the ideal part (Eqs. [1] and [2]) yields the full chemical potential $\mu$.

Below the critical temperature, the dependence of $\mu$ on $\rho$ contains a nonmonotonic part (known as the van der Waals loop) for finite-sized systems such as the protein solutions modeled here (see Fig. 3b). A horizontal line that bisects the van der Waals loop with equal areas below and above (equal-area construction) yields the concentrations of the protein in two coexisting phases. That is, the inner and outer intersections define the concentrations in the dilute and dense phases, respectively. By connecting the coexistence concentrations at a series of temperatures, one obtains the binodal. Moreover, the concentrations at the two extrema of the van der Waals loop define the bounds of a concentration range in which the system is thermodynamically unstable. By connecting the extremum concentrations at a series of temperatures, one obtains the spinodal.

Further details of polynomial fit and equal-area construction are found in ref. [32]. A web server implementing these steps is at https://zhougroup-uic.github.io/LLPS/.

**FMAPB2 calculations and related analyses.** The second virial coefficient $B_2$ is determined by the interaction energy between a pair of protein molecules. Let $U_1(\mathbf{R}, \boldsymbol{\Omega})$ be the pair interaction energy, then[30]

$$B_2 = -\frac{V}{2}\langle f(\mathbf{R}, \boldsymbol{\Omega})\rangle \qquad (20)$$

where $V$ is the volume in which the average over $\mathbf{R}$ is carried out, and $f(\mathbf{R}, \boldsymbol{\Omega})$ is the Mayer function:

$$f(\mathbf{R}, \boldsymbol{\Omega}) = e^{-\beta U_1(\mathbf{R}, \boldsymbol{\Omega})} - 1 \qquad (21)$$

We calculated $B_2$ also by an FFT-based method called FMAPB2[30]. The setup of FMAPB2 is identical to that of FMAPμ, but now the periodic box contains a single molecule, i.e., the central copy. The interaction energy function (see Eqs. [4] to [7]) and other details of FMAPB2 are the same as described above for FMAPμ calculations, with the following exceptions. The side length of the periodic box was ~200 Å. The test-protein orientations were generated from 68,760 rotation matrices that uniformly and deterministically sample the full orientation space, with a 6° angle between successive rotation matrices[57]. The large number of test-protein orientations makes the $B_2$ results highly robust (with a 3% difference in $\gamma$B $B_2$ at 25 °C when the test-protein orientation space was sampled at 4° intervals, involving 232,020 rotation matrices). In addition to the histogram of pair interaction energies, configurations with interaction energies lower than -6 kcal/mol were saved for further analysis.

We further analyzed the pair intermolecular interactions of the 1000 lowest-energy configurations using the atom-based method. Specifically, the pair interaction energy was decomposed into the contributions from the individual residues of the test protein interacting with the whole central copy. Because the test protein and the central copy are the same molecules, we also carried out the decomposition in the opposite way, i.e., with the central copy decomposed into individual residues and the test protein treated as a whole. For a given residue, the results were first averaged over the 1000 lowest-energy configurations and then over these two ways of decomposition.

We clustered the 1000 lowest-energy configurations by using the ligand-RMSD as the distance measure. Ligand-RMSD is the root-mean-square-deviation between two poses of the test protein after superposition on the central copy. We used a ligand-RMSD cutoff of 10 Å to define clusters. In each cluster, all members have ligand-RMSDs ≤ the cutoff from one member.

**Calculations with the Rosetta energy function and with side-chain repacking.** We used RosettaDock[47] (Rosetta version 3.13; https://www.rosettacommons.org/) on the 1000 lowest-energy poses from FMAPB2 calculations for $\gamma$B and $\gamma$F. We first wanted to see how the Rosetta all-atom energy function[46] may separate $\gamma$F from $\gamma$B in pair interaction energy. For this purpose, we applied a single round of energy minimization (option -dock_-min) and saved the resulting pair interaction energies.

We then wanted to see how sidechain repacking may affect the pair interaction energies. For this purpose, we carried out a refinement of the energy-minimized poses by preventing translation and rotation (movement magnitudes set to 0) but allowing sidechain repacking (option -use_input_sc and options -ex1 and -ex2aro, recommended by Rosetta). Up to ten tries of sidechain repacking were applied to obtain a negative interaction energy. If the interaction energies were positive in all the ten tries, the value from the last try was saved.

**Averaging $B_2$ over a preselected library of structures.** As another option to model protein flexibility in $B_2$ calculations, we allowed the central copy and the test protein to each sample a preselected library of $n$ structures, resulting in $n \times n$ pairs of structures. For each pair, a calculation was carried out using FMAPB23[39], which is the same as FMAPB2[30] except that the central copy and the test protein were allowed to take different structures. The final average $B_2$ was taken as the arithmetic mean of the $n \times n$ FMAPB23 results. For each particular $\gamma$-crystallin

(e.g., rat γE), the library consisted of all the high-resolution X-ray structures in the PDB; all the processed structures contained residues 1–174 (bovine γB numbering) and the same sequence.

**Grantham's distance between low and high-$T_c$ groups of γ-crystallins.** There are 12 sequences in the low-$T_c$ group and 3 sequences in the high-$T_c$ group (Supplementary Fig. 1a); correspondingly there are 66 Grantham distances in the low-$T_c$ group, 3 such distances in the high-$T_c$ group, and 36 such distances between the two groups, at each position along the sequence. We first calculated the mean values of the intra or inter-group distances, then subtracted the greater of the two intra-group mean distances from the inter-group mean distance, yielding the residual inter-group Grantham distance. Gln83 of γB aligns to a gap in 11 of the 15 sequences; we set the residual Grantham distance at this position to 0.

**Reporting summary.** Further information on research design is available in the Nature Portfolio Reporting Summary linked to this article.

## Data availability

All data generated or analyzed during this study are included in this published article (and its supplementary data files). The source data for all the plots presented in figures are deposited in GitHub at https://github.com/hzhou43/g-crystallins.

## Code availability

Data analysis procedures were described under Computational Methods. Sample codes for calculating chemical potentials by FMAP are deposited at https://github.com/hzhou43/MiMB_simulations/tree/main/FMAP. A web server for generating binodals from chemical potentials is at https://zhougroup-uic.github.io/LLPS/.

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

## Acknowledgements
This work was supported by National Institutes of Health Grant GM118091.

## Author contributions
S.Q. and H.X.Z. designed research, conducted research, analyzed data, and wrote manuscript.

## Competing interests
The authors declare no competing interests.
