## [Peer Review File · Communications Biology]

Reviewers' comments:

Reviewer #1 (Remarks to the Author):

In this intriguing manuscript, Qin and Zhou present atomistic simulations for calculating the phase diagram of liquid-liquid phase separation of γ -crystallins. Experimental observations have shown that γ B and γ F crystallins have different critical temperatures. For the first time, this study establishes a direct connection between the amino acid sequence and the experimentally observed difference in the proteins' critical temperatures. The authors rightfully argue that understanding this relationship is crucial for designing crystallins, where targeted mutations modify protein aggregates' phase diagrams and stability.

The connection between sequence and phase diagram is intriguing, and the presented methodology proves useful. By utilizing Fast Fourier Transforms, the authors efficiently analyze various experimental conditions, by varying chemical composition and temperature.

Nevertheless, this study does have weaknesses that suggest publishing it in a different venue. Following the conventional approach, the equilibrium between low-density and high-density solutions was determined by enforcing equality in chemical potential, pressure, and temperature in the two phases. The chemical potential is computed by inserting a protein into the crystallin matrix and measuring the change in internal energy. However, the protein insertion is performed using fully rigid structures, neglecting the consideration of changes in intramolecular energy upon insertion.

Furthermore, the procedure for calculating the internal energy, which is crucial for determining the chemical potential, is complex. The manuscript does not clarify the significance of the numerous computational approximations employed. It remains unclear whether certain arbitrary steps included in the procedure could account for the observed strong agreement with experimental results. A critical analysis of the adopted method should be included in this manuscript.

Interestingly, some of the weaknesses mentioned above are also acknowledged by the authors in their final discussion. They comment on the accuracy of the measured interaction energy, which depends on the procedure adopted. Additionally, they mention the absence of protein flexibility, which is a noteworthy limitation.

Reviewer #2 (Remarks to the Author):

Intracellular liquid-liquid phase separation (LLPS) has gained considerable interest in recent times due to its importance in many of the cellular processes vital to human health. In this work, the authors have used a state-of-the-art computational approach to model and study LLPS in γ -crystallines. This work is very timely and tries to establish a relationship between the protein's sequence and LLPS propensity. More precisely, this work tries to find the origin of the large (liquid-liquid) T_c difference between the highly homologous γ B and γ F proteins. This work can be of interest to others in the field of LLPS.

Although this manuscript is written very clearly, at few places the manuscript should be made more clear and the central result of the manuscript needs to have a stronger justification. For this reason, I recommend a major revision addressing the concerns below.

(1) The authors have reported that the substitution from Ser in γ B to Trp in γ F at position 130 is the major contributor to the observed T_c gap between these two homologous proteins. However, the authors have not explicitly reported the structural changes the protein would undergo after this mutation under experimental conditions (say, in water at ambient T and P). Any major structural change due to this mutation can alter the preferential interaction between the protein molecules, and in turn, T_c . As one of the major limitations of the approach used here is not taking into account the protein flexibility during the calculation of μ , it would be desirable to

know preferred protein conformations at experimentally relevant conditions before the start of the μ computation. The authors should report in the supplementary information the structure of a single protein molecule under experimental conditions (say, in water at ambient T and P) before and after mutation, and also comment on whether these structures are same as the ones used for the μ calculations.

(2) Page 7: the authors should highlight the basis for choosing the concentration range for γ_B and γ_F . Is this range experimentally relevant?

(3) Page 10, the authors reported that the γ_F Tc obtained in this study significantly underestimates the experimental values. Could the authors comment on the possible origin of this discrepancy?

Reviewer #3 (Remarks to the Author):

The authors use Brownian dynamics simulations to study the phase behaviour of two different sequences with an atomistic description. They mainly use Fast Fourier Transform (FFT) for computing chemical potentials by sampling configurations of the studied proteins. The authors claim that the difference in critical temperature between both sequences is due to a different amino acid at position 130 (Ser to Trp).

In the work, the authors explain the difference in critical point between the studied sequences by the substitution at position 130, claiming that Tryptophan interacts with higher binding strength than the Serine. The higher binding affinity of Trp is well-known by the community and the conclusions are what one can expect (see for instance *PNAS* **96**, 9459–9464 (1999); and *PLoS Computational Biology* **9**, e1003239 (2013)). However, the methods employed are valuable and thus can be useful for computational scientists studying phase separation of biomolecules. In that sense, I would recommend rewriting the manuscript changing the aim to focus more on the methodology rather than studying those specific proteins. Also, a contextualization and discussion of literature to the problem is lacking, so I would suggest extending the discussion of the methods presented as well as the number of references to provide a wider vision of the problem.

For these reasons I would recommend the editor to accept the manuscript after some major changes are made.

Major issues:

An important problem with this work is the contextualization and lack of literature review.

They can discuss the problem of altering protein sequence in experiments to alter phase behaviour (see *Nature Chemistry*, **14(2)**, 196-207 (2022) and *Cell* **174**, 688–699 (2018)). Also, the authors mention coarse-grained models in the abstract as limited to discuss residue effects on phase separation, but no discussion is included in the introduction. Indeed, there are recent models that are parameterized to reproduce sequence-dependent phase behaviour (see *PNAS* **117(46)**, 28795-28805 (2020); *PNAS* **118(44)**, e2111696118; *Nat.*

Computational Science, **1(11)**, 732-743 (2022)), and also there are works that studied the importance of different amino acid interactions in phase separating (*Biophysical Journal*, **120(23)**, 5169-5186 (2021)) or even avoiding structural transitions (bioRxiv 2022.12.14.520383). In addition, despite using atomistic models, the solvent is implicit. The authors should discuss the consequences and limitations of such modelling in contrast with other computational atomistic works (*PNAS* 119(26), e2119800119 (2021); *Nat. Comm.*, **12(1)**, 1085 (2021); *Nat. Comm.*, **13(1)**, 5717 (2022)).

To better understand the behaviour depicted in Fig. 2, I would suggest the authors to provide other observables to support what they state. Calculating the radius of gyration distribution or map of contacts may be useful, or even pair interaction. How to explain behaviour at increasing T? Also, why there's a downfall for the γ_B sequence around 280 mg/ml?

The phase diagram of γF critically underestimates the critical temperature of the experimental value. The authors point this out, but they don't provide any explanation or discussion regarding this significant difference. Can they discuss on this point? Could it be for underestimation of the electrostatic interactions due to the implicit solvent? In this sense, the authors conclude that the method they employ can be used to "design sequence with desired critical temperature". However, the model can't correctly predict the experimental critical point. I would suggest reducing this statement as the method can help to modify the phase behaviour in a certain direction rather than predict a desired T_c .

Figure S8 from the Supp. Material can be important to support the prediction of the phase behaviour from the estimation of the second virial term. I would suggest moving this figure to the main text but improving its visualization. In this sense, the difference between low and high critical temperature variants is significant when considering all the sequences, but it may be misleading if the main example of the paper are those PDBs that either underestimate or overestimate the critical point with the second virial term criterion.

As a work that attempts to use atomistic simulations to predict phase separation behaviour, it would greatly increase the adaptation of proposed best practices if authors could also share their data and code used to produce the results. A simple entry in Zenodo or Github with minimal documentation should suffice. Also, a minor comment in that respect, it would be useful to provide the protein sequence.

Minor comments:

- Line 35 of the manuscript should be "the fact that" .
- In Figure 1 it may be useful to indicate which is the colour of each protein variant in panel B as well.
- In line 122, I don't see how Fig. 2A is related to the statement before. This need to be clarified.
- The abbreviation for millilitres is ml instead of mL. Check Figs. 2-4, line 797,, 804

We thank the reviewers for their constructive comments. Our point-by-point response is given below in blue.

Reviewer #1 (Remarks to the Author):

In this intriguing manuscript, Qin and Zhou present atomistic simulations for calculating the phase diagram of liquid-liquid phase separation of γ -crystallins. Experimental observations have shown that γ B and γ F crystallins have different critical temperatures. For the first time, this study establishes a direct connection between the amino acid sequence and the experimentally observed difference in the proteins' critical temperatures. The authors rightfully argue that understanding this relationship is crucial for designing crystallins, where targeted mutations modify protein aggregates' phase diagrams and stability.

The connection between sequence and phase diagram is intriguing, and the presented methodology proves useful. By utilizing Fast Fourier Transforms, the authors efficiently analyze various experimental conditions, by varying chemical composition and temperature.

Nevertheless, this study does have weaknesses that suggest publishing it in a different venue. Following the conventional approach, the equilibrium between low-density and high-density solutions was determined by enforcing equality in chemical potential, pressure, and temperature in the two phases. The chemical potential is computed by inserting a protein into the crystallin matrix and measuring the change in internal energy. However, the protein insertion is performed using fully rigid structures, neglecting the consideration of changes in intramolecular energy upon insertion.

We address the limitation of using rigid structures in two ways. First off, in the new Supplementary Note 1, we discuss the choices we made to make the calculations feasible. In essence, the compromise necessitated by feasibility is either reducing the representation of proteins from all-atom to coarse-grained, or treating the proteins as rigid. For our task of characterizing the sequence dependence of the binodals of structured proteins like γ -crystallins, it is more important to retain an all-atom representation than to treat flexibility. Moreover, we now report preliminary results from accounting for protein flexibility (new Figs. S10 and S11). These results demonstrate that treating flexibility improves accuracy and robust, but the main conclusions remain the same.

Furthermore, the procedure for calculating the internal energy, which is crucial for determining the chemical potential, is complex. The manuscript does not clarify the significance of the numerous computational approximations employed. It remains unclear whether certain arbitrary steps included in the procedure could account for the observed strong agreement with experimental results. A critical analysis of the adopted method should be included in this manuscript.

We appreciate the reviewer's recognition of the "strong agreement" between our calculations and experimental results. The central part of the procedure for calculating the interaction energy involves FFT, which has been carefully validated in a previous publication (ref. 34). Other components have likewise been thoroughly benchmarked – there are no "arbitrary steps". We

now include a critical analysis of the adopted method (new Supplementary Note 1). We also present additional results showing how the parameterization of our interaction energy function may affect the results (new Fig. S7). Again, the main conclusions remain the same.

Interestingly, some of the weaknesses mentioned above are also acknowledged by the authors in their final discussion. They comment on the accuracy of the measured interaction energy, which depends on the procedure adopted. Additionally, they mention the absence of protein flexibility, which is a noteworthy limitation.

We have indeed been very open about the two main weaknesses of our approach: limitation of implicit-solvent energy function and lack of flexibility. We now add additional calculations, to further demonstrate the robustness of our conclusions (new Figs. S7, S10, and S11).

Reviewer #2 (Remarks to the Author):

Intracellular liquid-liquid phase separation (LLPS) has gained considerable interest in recent times due to its importance in many of the cellular processes vital to human health. In this work, the authors have used a state-of-the-art computational approach to model and study LLPS in γ -crystallines. This work is very timely and tries to establish a relationship between the protein's sequence and LLPS propensity. More precisely, this work tries to find the origin of the large (liquid-liquid) T_c difference between the highly homologous γ_B and γ_F proteins. This work can be of interest to others in the field of LLPS.

Although this manuscript is written very clearly, at few places the manuscript should be made more clear and the central result of the manuscript needs to have a stronger justification. For this reason, I recommend a major revision addressing the concerns below.

(1) The authors have reported that the substitution from Ser in γ_B to Trp in γ_F at position 130 is the major contributor to the observed T_c gap between these two homologous proteins. However, the authors have not explicitly reported the structural changes the protein would undergo after this mutation under experimental conditions (say, in water at ambient T and P). Any major structural change due to this mutation can alter the preferential interaction between the protein molecules, and in turn, T_c . As one of the major limitations of the approach used here is not taking into account the protein flexibility during the calculation of μ , it would be desirable to know preferred protein conformations at experimentally relevant conditions before the start of the μ computation. The authors should report in the supplementary information the structure of a single protein molecule under experimental conditions (say, in water at ambient T and P) before and after mutation, and also comment on whether these structures are same as the ones used for the μ calculations.

We actually present in Fig. 1b the crystal structures of both γ_B and γ_F , showing high similarity and without any major changes. Indeed, the more than two dozen high-resolution crystal structures of mammalian γ -crystallins in the Protein Data Bank all show high similarities. Also note that we calculated the chemical potentials of γ_B and γ_F , using the structures shown in Fig. 1b. As a corroborative study, we did additional calculations on point mutants, where Ser130 in γ_B

was mutated into Trp and Trp130 in γ F was mutated into Ser. The structures of these mutants were modeled by changing only the mutated residues, based on foregoing preponderance of evidence for structure conservation.

(2) Page 7: the authors should highlight the basis for choosing the concentration range for γ B and γ F. Is this range experimentally relevant?

Yes, we now explicitly state that the range of concentrations in our calculations is the experimentally relevant range (p. 7).

(3) Page 10, the authors reported that the γ F Tc obtained in this study significantly underestimates the experimental values. Could the authors comment on the possible origin of this discrepancy?

We now explicitly state that the origins of this discrepancy are most likely the two main limitations of our approach, i.e., implicit-solvent energy function and lack of flexibility (p. 17). We also present preliminary results that indeed suggest that a more sophisticated energy function and inclusion of protein flexibility could increase the γ F Tc (Fig. S10).

Reviewer #3 (Remarks to the Author):

The authors use Brownian dynamics simulations to study the phase behaviour of two different sequences with an atomistic description. They mainly use Fast Fourier Transform (FFT) for computing chemical potentials by sampling configurations of the studied proteins. The authors claim that the difference in critical temperature between both sequences is due to a different amino acid at position 130 (Ser to Trp).

In the work, the authors explain the difference in critical point between the studied sequences by the substitution at position 130, claiming that Tryptophan interacts with higher binding strength than the Serine. The higher binding affinity of Trp is well-known by the community and the conclusions are what one can expect (see for instance *PNAS* **96**, 9459– 9464 (1999); and *PLoS Computational Biology* **9**, e1003239 (2013)). However, the methods employed are valuable and thus can be useful for computational scientists studying phase separation of biomolecules. In that sense, I would recommend rewriting the manuscript changing the aim to focus more on the methodology rather than studying those specific proteins. Also, a contextualization and discussion of literature to the problem is lacking, so I would suggest extending the discussion of the methods presented as well as the number of references to provide a wider vision of the problem.

We now add an extended discussion of the methodology to provide a proper context (new Supplementary Note 1 and new Fig. S2). We also cite the two mentioned papers for the roles of Trp in folding and binding stability (p. 13).

For these reasons I would recommend the editor to accept the manuscript after some major changes are made.

Major issues:

An important problem with this work is the contextualization and lack of literature review. They can discuss the problem of altering protein sequence in experiments to alter phase behaviour (see *Nature Chemistry*, **14**(2), 196-207 (2022) and *Cell* **174**, 688–699 (2018)). Also, the authors mention coarse-grained models in the abstract as limited to discuss residue effects on phase separation, but no discussion is included in the introduction. Indeed, there are recent models that are parameterized to reproduce sequence-dependent phase behaviour (see *PNAS* **117**(46), 28795-28805 (2020); *PNAS* **118**(44), e2111696118; *Nat. Computational Science*, **1**(11), 732-743 (2022)), and also there are works that studied the importance of different amino acid interactions in phase separating (*Biophysical Journal*, **120**(23), 5169-5186 (2021)) or even avoiding structural transitions (bioRxiv 2022.12.14.520383). In addition, despite using atomistic models, the solvent is implicit. The authors should discuss the consequences and limitations of such modelling in contrast with other computational atomistic works (*PNAS* 119(26), e2119800119 (2021); *Nat. Comm.*, **12**(1), 1085 (2021); *Nat. Comm.*, **13**(1), 5717 (2022)).

We already cited *Cell* **174**, 688–699 (2018) (ref. 5) and a predecessor (ref. 7) of *Nature Chemistry*, **14**(2), 196-207 (2022), and have now added citation to the latter paper (ref. 10). We now also cite the other noted papers in the main text (p. 5) and in the new Supplementary Note 1.

To better understand the behaviour depicted in Fig. 2, I would suggest the authors to provide other observables to support what they state. Calculating the radius of gyration distribution or map of contacts may be useful, or even pair interaction. How to explain behaviour at increasing T? Also, why there's a downfall for the γB sequence around 280 mg/ml?

The reviewer's point is well taken, but we do present other observables to support our conclusion. In particular, we report pair interaction results in Figs. 4a, 5, and S8, and display contacts in Figs. 6 and S9. We also state that "As the temperature is raised, steric interactions become important even at lower protein concentrations, thereby elevating the entire μ_{ex} curve" (p. 8, fourth paragraph). In essence, while soft interactions are weighted down by $k_B T$, steric interactions are independent of temperature and hence become dominant at high temperatures. As for the μ value of γB at 277 mg/ml and -2 °C, we have already pointed out that this excessively low value is due to the difficulty of sampling at low temperatures (p. 8, third paragraph). We have developed a scheme to remove the effects of such outliers (p. 22-24 in Methods section).

The phase diagram of γF critically underestimates the critical temperature of the experimental value. The authors point this out, but they don't provide any explanation or discussion regarding this significant difference. Can they discuss on this point? Could it be for underestimation of the electrostatic interactions due to the implicit solvent? In this sense, the authors conclude that the method they employ can be used to "design sequence with desired critical temperature". However, the model can't correctly predict the experimental critical point. I would suggest reducing this statement as the method can help to modify the phase behaviour in a certain direction rather than predict a desired T_c .

We now explicitly state that the origins of this discrepancy are most likely the two main limitations of our approach, i.e., implicit-solvent energy function and lack of flexibility (p. 17). We also present evidence suggesting that a more sophisticated energy function and inclusion of

protein flexibility increase the $\gamma^F T_c$ (Fig. S10). Lastly, we weakened the statement about sequence design (p. 16).

Figure S8 from the Supp. Material can be important to support the prediction of the phase behaviour from the estimation of the second virial term. I would suggest moving this figure to the main text but improving its visualization. In this sense, the difference between low and high critical temperature variants is significant when considering all the sequences, but it may be misleading if the main example of the paper are those PDBs that either underestimate or overestimate the critical point with the second virial term criterion.

This is an excellent suggestion! We have now moved the original Fig. S8 to the main text as Fig. 7, and improved its appearance.

As a work that attempts to use atomistic simulations to predict phase separation behaviour, it would greatly increase the adaptation of proposed best practices if authors could also share their data and code used to produce the results. A simple entry in Zenodo or Github with minimal documentation should suffice. Also, a minor comment in that respect, it would be useful to provide the protein sequence.

We have deposited the data and codes in GitHub, as described in our Data availability and Code availability statements. Also, the sequences as well as the UniProt entry names of all the proteins studied are given in Fig. S1.

Minor comments:

- Line 35 of the manuscript should be “the fact that” .

Corrected

- In Figure 1 it may be useful to indicate which is the colour of each protein variant in panel B as well.

Good point; done

- In line 122, I don't see how Fig. 2A is related to the statement before. This needs to be clarified.

We have clarified this point by adding explanations in p. 6 and in the Fig. 2a caption, and by modifying Fig. 2a slightly.

- The abbreviation for millilitres is ml instead of mL. Check Figs. 2-4, line 797,, 804

Corrected

REVIEWERS' COMMENTS:

Reviewer #3 (Remarks to the Author):

The authors have properly addressed the previous review comments. In my opinion, the manuscript is ready for publication.